# Determination of Cultivation Regions and Quality Parameters of *Poria cocos* by Near-Infrared Spectroscopy and Chemometrics

**DOI:** 10.3390/foods11060892

**Published:** 2022-03-21

**Authors:** Jing Xie, Jianhua Huang, Guangxi Ren, Jian Jin, Lin Chen, Can Zhong, Yuan Cai, Hao Liu, Rongrong Zhou, Yuhui Qin, Shuihan Zhang

**Affiliations:** 1Hunan Academy of Chinese Medicine, Hunan University of Chinese Medicine, Changsha 410013, China; axxj2057@163.com (J.X.); jhhuang85@163.com (J.H.); jinjian2016@163.com (J.J.); chenlin5202@126.com (L.C.); canzhong651@163.com (C.Z.); tcmyuanyuan@163.com (Y.C.); zrziqing@126.com (H.L.); rz172@georgetown.edu (R.Z.); hntcmaqyh@126.com (Y.Q.); 2College of Chinese Pharmacy, Beijing University of Chinese Medicine, Beijing 102488, China; renguangxiabc@163.com

**Keywords:** *Poria cocos*, NIR spectroscopy, chemometric methods, chemical compositions, cultivation regions

## Abstract

*Poria cocos* (PC) is an important fungus with high medicinal and nutritional values. However, the quality of PC is heavily dependent on multiple factors in the cultivation regions. Traditional methods are not able to perform quality evaluation for this fungus in a short time, and a new method is needed for rapid quality assessment. Here, we used near-infrared (NIR) spectroscopy combined with chemometric method to identify the cultivation regions and determine PC chemical compositions. In our study, 138 batches of samples were collected and their cultivation regions were distinguished by combining NIR spectroscopy and random forest method (RFM) with an accuracy as high as 92.59%. In the meantime, we used partial least square regression (PLSR) to build quantitative models and measure the content of water-soluble extract (WSE), ethanol-soluble extract (ASE), polysaccharides (PSC) and the sum of five triterpenoids (SFT). The performance of these models were verified with correlation coefficients (*R*^2^_cal_ and *R*^2^_pre_) above 0.9 for the four quality parameters and the relative errors (RE) of PSC, WSE, ASE and SFT at 4.055%, 3.821%, 4.344% and 3.744%, respectively. Overall, a new approach was developed and validated which is able to distinguish PC production regions, quantify its chemical contents, and effectively evaluate PC quality.

## 1. Introduction

*Poria cocos* (PC) is derived from dried sclerotium of the fungus *P. cocos* (Schw.) Wolf, and it provides important nutritional and medicinal values. In China, it is known as “Fuling” and used for the treatment of various diseases [1], such as diabetes [2], cancer [3,4] and depression [5]. PC usage for nutritional and medicinal purposes has a long history of over 2000 years. PC was first recorded in an ancient Chinese medical book, Wushi’er Bingfang or Recipes for Fifty-Two Ailments, which described its clinical efficacy in promoting urination, removing dampness, invigorating the spleen and calming nerves [6,7,8]. Recent reports demonstrated that recipes with PC have been developed as part of a daily diet to prevent various chronic diseases such as hyperlipidemia, hyperglycemia, hypertension and atherosclerosis [7,8]. In China, these recipes are often used to develop and produce popular functional foods for both nutritional and medical benefits.

Since PC offers remarkable medicinal values with little side effects, the commercial PC market has grown rapidly in recent years. Cultivated PC has grown strongly due to the high demand for it that cannot be met by very limited amount of wild-growing PC. PC is a Chinese medicinal material, and similar to most materials for Chinese medicine, its quality is dependent on multiple factors within the cultivation region, including planting ecological environments [9], harvesting time [10], processing technology [11], etc. These factors affect the generation of secondary metabolites, which are important indicators of the quality and nutritional value of medicinal materials, including PC. 

PC is mainly distributed in East and Southeast Asia where the climate is humid and subtropical, including China, Vietnam, and the Philippines [12,13]. The Dabie Mountains region, Yunnan region and Xiangqian region are the three main cultivation regions of PC in China, and the products of these regions, which are across different climate zones, have their own features [10,11,14,15,16]. The diverse cultivation conditions in these remotely distanced regions have caused quality discrepancies in PC grown in China. Identification of PC’s cultivation region in China would greatly benefit its quality classification, which is part of product quality control. This can be accomplished by developing a quick and accurate method to measure the levels of some key constituents of PC that are dependent on the cultivation region.

According to the Chinese and European Pharmacopoeia, the contents of water-soluble extracts and ethanol-soluble extracts are important indexes for the quality evaluation of PC. These contents contain a variety of chemical constituents, including triterpenes, polysaccharides, sterols, essential oil, proteins, etc. [13]. Among them, the main bioactive constituents are polysaccharides and triterpenes, which have been reported for various pharmacological benefits such as anti-inflammatory, antibacterial, antitumor and antioxidant effects. These pharmacological benefits contribute to the clinical efficacy of PC [17,18,19,20]. PC polysaccharides are a mixture of different types of polysaccharide, and account for 70~89% of the total dried mass of dried sclerotium, but only 2% of it dissolves in water—these water-soluble polysaccharides are the foundation of treating diseases, so PC is generally taken in the form of a water decoction [3]. PC polysaccharides contains rich branched glucans with (1,3)-β,(1,4)-β and (1,6)-β linkages and has exhibited strong bioactivity [21,22,23,24,25]. Up to now, more than 91 bioactive triterpenes have been isolated [13] and identified from PC, including Dehydrotumulosic acid (DTUA), Poricoic acid A (PAA), Polyporenic acid C (PAC), Dehydropachymic acid (DPA), Dehydrotrametenolic acid (DTRA) and other active triterpenoids [26,27]. Among them, DTRA and PAC are lanostane-type triterpenes, DPA and DTUA are Eburicane-type triterpenes, PAA is seco-Eburicane-type triterpenes, and they all show promising bioactivity [28,29]. These chemical constituents have special absorption spectra that show a unique reflection of their composition and molecular structure. This feature can correlate with the spectral response of sample to its constituents [30]. 

A broad range of analytical methods are available to assess the content of these chemicals in PC. Among them, near-infrared (NIR) spectroscopy is an advancing analytical technique that is rapid, easy to use and nondestructive [31,32]. NIR has been widely used to evaluate the quality of edible fungi, such as *Pleurotus ostreatus* [33], *Ganoderma lucidum* [34] and some cultivated mushrooms [35,36], with the capability to classify and identify cultivation regions [37]. NIR provides a complex spectrum with comprehensive absorption peak of H–containing groups (i.e., N–H, O–H, and C–H) that can be found in many active substances, including polysaccharides, flavonoids, triterpenes and phenols, etc. However, the spectroscopy of these substances is very sensitive to external interferences introduced by variances in ambient temperature, light scattering of the spectrometer and the color and physical status of samples. The raw spectra of these substances are similar and broad, consisting of many overlapping narrow bands, and hence make it challenging for further data analysis. To overcome this challenge, some spectral processing methods, such as standard normal variate (SNV) [38], smoothing, multiplicative scanter correction (MSC) [39] and Savizaky–Golay method (SG) [40,41], have to be used to optimize the calibration model by coupling with chemometrics. As previously stated, it is difficult to thoroughly assess the quality of Chinese medicinal herbs, due to the complexity of their active ingredients. However, NIR technology, combined with chemometric techniques, provides new insights into quality assessment of Chinese medicine materials with improved accuracy, and can be used for comprehensive analysis of the chemical composition of PC.

The objective of this study is to establish new models to distinguish cultivation regions and quantify the chemical composition of PC using NIR and chemometrics. Firstly, 138 batches of PC samples were collected from the three main cultivation regions, and then their raw NIR spectra and the content of water-soluble extract, alcohol-soluble extract, polysaccharides and the five triterpene acids were measured. Secondly, we used random forest method (RFM) to distinguish cultivation regions by NIR spectroscopy, and then used partial least square regression (PLSR) method to develop calibration model to determine the concentration of water-soluble extract, alcohol-soluble extract, polysaccharides and the sum of the five triterpenoids.

## 2. Materials and Methods

### 2.1. Sample Preparation

A total of 138 batches of PC samples were collected from the three major PC cultivation regions (Dabie Mountains region, Yunnan region and Xiangqian region). Detailed information of these samples is listed in Appendix A. These samples were identified and authenticated by associate Prof. Hao Liu of Hunan Academy of Chinese Medicine. All the dry samples were further dried in an oven at 60 °C for 2 h, ground and sieved through a 60 mesh (0.3 mm). Well-prepared powder samples were labeled and stored in a dark and dry place.

### 2.2. Chemicals and Reagents

D-glucose (≥98.0% purity) was purchased from Sigma-Aldrich (Sigma, St. Louis, MA, USA). Dehydrotumulosic acid (DTUA, CAS No. 6754-16-1), poricoic acid A (PAA, CAS No. 137551-38-3), polyporenic acid C (PAC, CAS No. 465-18-9), dehydropachymic acid (DPA, CAS No. 77012-31-8), DehydrotraMetenolic acid (DTRA, CAS No. 29220-16-4), all above 98% purity, were purchased from Chroma Biotechnology Co., Ltd. (Chengdu, China). Distilled water was purified using a Milli-Q system (Millipore, Bedford, MA, USA). Acetonitrile (HPLC-grade) was purchased from Merck (Darmstadt, Germany). Ultrapure water was purchased from Hangzhou Wahaha Group Co., Ltd. (Changsha, China). All the reagents and chemicals were of analytical grade.

### 2.3. Extraction of Water-Soluble Content and Ethanol-Soluble Content of PC

The water-soluble and ethanol-soluble contents of the solutions were then extracted and analyzed by following the standard method documented in the Chinese Pharmacopoeia Volume IV: 2201 (2020 Edition).

To determine the content of water-soluble extract: 2.5 g sample powder was mixed in 50 mL water to make a water solution, which was weighed and allowed to stand for 1 h. The solution was then heated in a water bath to slightly boil for 1 h. After this, the solution was cooled and its weight was replenished by adding water. The well-mixed water solution was then filtered and 25 mL of the filtrate was transferred to an evaporating dish, which was heated in a water bath to obtain the extract by removing water through evaporation. The extract was dried at 105 °C for 3 h and allowed to cool for 30 min in a desiccator. Immediately after that, the extract was accurately weighted. The water-soluble extract content was determined using the following equation:*C_w_* = (*W*_1_ − *W*_0_) × 2/*S* × 100%(1)
where *Cw* is the water-soluble extract content (%), *S* is the mass of the sample (g), *W*_1_ is the mass of the evaporating dish and residue after drying (g), and *W*_0_ is the mass of the evaporating dish (g).

The content of ethanol-soluble extract was determined in a similar way by using ethanol instead of water.

### 2.4. Polysaccharides Content and Absorbance Curve

The contents of PC samples were determined using the phenol-sulphuric acid method. The powder samples (0.5 g) were mixed with 100 mL of 80% (*v*/*v*) ethanol, and the solution was heated in a water bath to reflux for 1 h. The solution was then filtered through a hopper to obtain the residue that was then washed with 15 mL of 80% (*v*/*v*) hot ethanol for three times. The remaining residue was heated in 75 mL water and refluxed for 4 h. The residue solution was then filtered while it was hot and the filter was washed with a small amount of hot water. The filtrate was diluted with water in a 100 mL flask. In total, 1 mL of filtrate was precisely measured, placed in a 10 mL test tube, and further homogenized with 0.8 mL of 5% (*v*/*v*) phenol and 3.5 mL of concentrated sulphuric acid. The mixture was incubated in water bath at 40 °C for 15 min. After each sample was cooled in an ice bath, its absorbance was measured at a wavelength of 488 nm. A total of 0.098 mg/mL standard solution of glucose was prepared as reference solution. The sample was diluted for different concentrations of 0.0018, 0.0037, 0.0074, 0.0111, 0.0148 and 0.0185 mg/mL. The D-glucose standard was used to generate the standard absorbance curve, with the equation Y = 84.978X + 0.0571, *R*^2^ = 0.9924 (X—sample concentration; Y—absorbance).

### 2.5. Quantitative Analysis of Five Triterpene Acids Using High-Performance Liquid Chromatography (HPLC)

Chromatographic analysis was performed using a HPLC system (1200 series, Agilent Technologies, Santa Clara, CA, USA), comprising a quaternary solvent delivery pump, online degasser, autosampler, column thermostat and diode array detector. The column (ZORBAX SB-C18 column, 5 μm, 4.6 × 250 mm) was performed at 30 °C and at the flow rate of 1.0 mL/min. Acetonitrile and 0.2% (*v*/*v*) formic acid/water were used as mobile phase A and B, respectively. The solvent gradient was set as follows: 51% A during 0–5 min, 51–57% A during 5–20 min, 57–70% A during 20–45 min, 70–85% A during 45–55 min, and 85–100% A during 55–60 min. The injection volume was 10 μL and the detection wavelength was 245 nm.

Powdered samples (2.0 g) were accurately weighed, and then extracted with 20 mL methanol by ultrasonication (KM500DB, Kunshan Meimei Ultrasonic Instrument Co., Ltd., Kunshan, Jiangsu, China) at 40 kHz for 30 min. The extracts were then passed through 0.45 μm syringe filter for HPLC analysis. The concentrations of five triterpene acids, including DTUA, PAA, PAC, DPA and DTRA, were determined using standard calibration curves, and the detailed information of standard calibration curves was listed in Table 1.

### 2.6. NIR Spectroscopy Determination

The NIR spectra of the samples were measured by Antaris II FT-NIR spectrometer (Thermo Electron Co., Waltham, MA, USA), and the parameters of spectra were determined as 32 iterations of scans, resolution 8 cm^−1^, and scan range 600 to 2000 nm. The powder sample was added to the sample cup and compressed. Each sample was measured three times and the average of three spectra was calculated for further statistical analysis. The temperature was kept at around 25 °C in the laboratory.

### 2.7. Chemometric Methods

#### 2.7.1. Geographical Discriminant Analysis of Random Forest Method (RFM)

RFM was an integration algorithm based on classification regression tree [42]. We used it to distinguish the producing areas of PC samples. During the training process, the RFM method constructed multiple independent tree classifiers based on Bagging algorithm, and the final results were voted by all constructed training classifiers. After model training, it can be used to classify the samples according to their producing areas. Previous studies [43] had reported in more detail how RFM performed better in distinguishing producing areas of Chinese medicinal materials.

#### 2.7.2. Partial Least Squares Regression (PLSR) and Determination of Chemical Composition

PLSR is a classical regression method and has been widely used for quantitative determination through modeling. PLSR intends to project the high-dimensional predictor variables into a smart set of latent variables, which have a maximal covariance to the responses [43]. In this study, we used PLSR to establish the regression model for determination of chemical composition content, i.e., polysaccharides, ethanol-soluble extract, water-soluble extract and the five triterpenoids. The regression model was able to learn the relationship between independent variable X (NIR matrix) and dependent variable Y (chemical composition content), and the determination of chemical composition content was achieved by multiple linear regression of score matrix.

### 2.8. Software Requirements and Data Analysis

Statistical evaluation was performed using SPSS 21.0 (SPSS Inc., Chicago, IL, USA). All the algorithms were implemented in MATLAB 8.0.1 (MathWorks), and all the programing was done by the authors.

The indicators for evaluating the calibration set were the root mean square error of calibration (RMSEC), the root mean square error of prediction (RMSEP), the coefficient of determination for calibration (*R*^2^_cal_) and the coefficient of determination for prediction (*R*^2^_pre_). RMSEC and RMSEP were used to measure the model effectiveness and accuracy, while *R*^2^_cal_ and *R*^2^_pre_ were used to evaluate the fit of model. More specifically, a better model is indicated by smaller values of RMSEC and RMSEP and *R*^2^_cal_ and *R*^2^_pre_ being closer to 1. RMSEC, RMSEP and *R*^2^ were respectively defined using the following equation:(2)$$\mathrm{RMSEC}=\sqrt{\frac{1}{n_{c}}\sum_{i=1}^{n_{c}}{\left({ŷ}_{i}-y_{i}\right)}^{2}}$$
(3)$$\mathrm{RMSEP}=\sqrt{\frac{1}{n_{p}}\sum_{i=1}^{n_{p}}{\left({ŷ}_{i}-y_{i}\right)}^{2}}$$
(4)$$R^{2}{\mathrm{cal}}_{\;}=1-\frac{\sum_{i=1}^{n_{c}}{\left({ŷ}_{i}-y_{i}\right)}^{2}}{\sum_{i=1}^{n_{c}}{\left(y_{i}-\;ȳ\right)}^{2}}$$
(5)$$R^{2}{\mathrm{pre}}_{\;}=1-\frac{\sum_{i=1}^{n_{p}}{\left({ŷ}_{i}-y_{i}\right)}^{2}}{\sum_{i=1}^{n_{p}}{\left(y_{i}-\;ȳ\right)}^{2}}\;$$
where $n_{c}$, $n_{p}$ ${ŷ}_{i}$, $y_{i}$ and ȳ are the number of observations in calibration, number of observations in prediction set, predicted value of the i-th observation, measured value of i-th observation and average of measured values, respectively.

## 3. Results and Discussion

### 3.1. Chemical Composition of PC Samples

#### 3.1.1. Polysaccharides, Water-Soluble Extract and Ethanol-Soluble Extract in PC Samples

According to Chinese, Japanese, Korean and European pharmacopoeias, ethanol-soluble extract, water-soluble extract and polysaccharides are important indicators to evaluate the quality of PC. Our study had detected all these three indicators, and the detailed contents for these chemical compositions are listed in Table 2 and Figure 1A–C. Samples from the Yunnan region have the highest content of polysaccharide, water-soluble extract and ethanol-soluble extract at 6.81 mg g^−1^, 25.23 mg g^−1^ and 37.85 mg g^−1^, respectively. Samples from the Dabie Mountains region contain the lowest levels of polysaccharide (4.59 mg g^−1^), water-soluble extract (19.57 mg g^−1^) and ethanol-soluble extract (31.24 mg g^−1^). Figure 1A–C shows that the polysaccharides, ethanol-soluble extract, and water-soluble extract from the Yunnan region were significantly (*p* < 0.05) higher than those from the Dabie Mountains region. The contents of polysaccharides, water-soluble extract and ethanol-soluble extract in all the samples ranged from 1.24 mg g^−1^ to 31.16 mg g^−1^, 8.80 mg g^−1^ to 73.95 mg g^−1^, and 16.29 mg g^−1^ to 90.00 mg g^−1^, respectively. The wide range of these values demonstrates that the quality of PC varies greatly among different producing regions. With regard to the standard deviation, the Yunnan region is highest in all of the three chemical compositions, followed by the Xiangqian region and the Dabie Mountains region. Although samples from the Yunnan region have high quality, samples from the Dabie Mountains region show better consistency in quality. The high quality of PC products in the Yunnan region could be attributed to the local climate and natural environment, while the quality consistency in samples from the Dabie Mountains region could be related to higher standard of planting and processing.

#### 3.1.2. HPLC Analysis of Five Triterpene Acids

Triterpenes is also an important indicator of the quality of PC. In this study, DTUA, PAA, PAC, DPA and DTRA were selected as the leading indicators and the contents of the five triterpene acids in these PC samples were determined via HPLC analysis. The detailed results were listed in Table 2 and plotted in Figure 1D–I. The results show that the contents of DTUA, PAC and DPA in samples from the Dabie Mountains region are significantly higher than those from the Yunnan region, while the contents of PAA and DTRA are slightly lower than those from the Yunnan region. However, the total contents of the five triterpenoids in the samples do not change significantly from one cultivation region to another. Similar to polysaccharides, ethanol-soluble extract and water-soluble extract, contents of triterpenoids vary greatly among the producing regions. The result also suggests that high contents of five triterpenoids in PC corresponds to low contents of polysaccharides, and vice versa.

### 3.2. NIR Spectra Characteristics

The samples were analyzed using NIR spectroscopy in the wavenumber range of 12,000 to 4000 cm^−1^. The results are presented in Figure 2, which illustrates five absorption peaks were mainly concentrated at about ~8256, ~6880, ~5610, ~5180 and ~4800 cm^−1^. The chemical structures of PC chemical components have abundant C–H and O–H bonds. It can be considered that the chemical components of PC were responsible for the NIR spectra. The strongest absorbance peak was observed at the wavenumber of ~5180–5200 cm^−1^, which is related to the combination of O–H stretching and bending, while the peak at ~4800 cm^−1^ is related to the combination of both C–O stretching and O–H deformation. The peak at ~6880 cm^−1^ may be related to the first overtone absorption of O–H stretching vibration and the weak peak at ~8256 cm^−1^ is related to the second overtone of symmetric stretching of the –CH bonds of –CH_3_ groups. Other less evident absorbance peaks are caused by overtones and combination modes of C–H, N–H, and O–H. The O–H and C–H vibrations are caused by compounds such as polysaccharides, triterpenoids and proteins [34,44]. As shown in Figure 2, the high overlap of NIR spectra makes it difficult to directly utilize the spectral information of target components. This problem can be solved by preprocessing the spectral information for better signal-to-noise ratio.

### 3.3. Elimination of Outliers

Outliers can have deleterious effects on model accuracy. We used Monte Carlo cross-validation method to identify potential outliers in NIR spectra. As shown in Figure 3, three samples were identified as outliers and thus excluded from model building.

### 3.4. Determination of PC Cultivation Regions

A classification model based on an RFM integration algorithm was used to identify PC cultivation regions. RFM integration algorithm could classify samples from different cultivation regions based on their similarities calculated in the model-training process.

To develop a more reliable model, samples were divided into calibration group (108 samples) and validation group (27 samples) at a ratio of 4:1 by classical Kennard–Stone (K-S) algorithm. The calibration group was used to optimize the model parameters and establish the optimal classification model. The validation group was used to verify the performance of the classification model. The separation plot of samples from different cultivation regions was shown in Figure 4.

Furthermore, the validation group was used to validate the reliability of the classification model we developed. The accuracy of classification prediction model was calculated as 92.59%, or 25 out of 27, with false results from only two samples (Table 3).

### 3.5. Development of Quantitative Prediction Model

#### 3.5.1. Model Development and Optimization

By using Kennard–Stone (K-S) algorithm, 108 samples were selected to optimize the pretreatment processes and develop the prediction model, while the remaining 27 samples were used to validate the quantitative prediction model. Detailed information is provided in Table 4; the results show that the calibration group covers the range of the validation group, making the model more stable. In this section, a PLSR algorithm was used to build these quantitative prediction models, and the number of PLSR factors were optimized by 10-fold cross-validation method.

We further employed several signal pretreatment methods, including multiplicative scatter correction (MSC), Savizky–Golay method (SG), standard normal variate (SNV), and combined process methods, to improve the model performance, which was measured by RMSEC, *R*^2^_cal_, RMSEP and *R*^2^_pre_. The results obtained by different signal preprocess methods were showed in Table 5 and suggested that SG-1D was the most suitable for the water-soluble extracts and polysaccharides, while SNV worked best for the ethanol-soluble extracts and sum of five triterpenoids.

The results in Table 5 allow us to develop the prediction models using the optimized experimental conditions. The model for determination of water-soluble extract, alcohol-soluble extract, polysaccharides and the sum of five triterpenoids generated *R*^2^_cal_ of 0.978, 0.972, 0.987 and 0.975, respectively, and exhibited RMSEC values of 1.083, 1.502, 0.056 and 0.165, respectively. In general, a good model would generate results with *R*^2^_cal_ and *R*^2^_pre_ close to 1, and RMSEC and RMSEP close to 0. A value of *R* greater than 0.9 indicates a strong correlation between the determined and actual values, and great accuracy of determination. The results in Table 5, with *R*_cal_ and *R*_pre_ values close to 1 and RMSEP and RMSEC values close to 0, demonstrate that the calibration models are capable of identifying the producing regions with good accuracy.

#### 3.5.2. Validation of Quantitative Model

After the above processes were completed, the validation group of samples were used to validate the optimal prediction model. From Table 5, the values of *R*^2^_pre_ and RMSEP are: 0.962 and 1.934, respectively, for water-soluble extract; 0.940 and 1.926, respectively, for alcohol-soluble extract; 0.965 and 0.0079, respectively, for polysaccharides; and 0.961 and 0.191, respectively, for the sum of five triterpenoids, as listed in Table 5. These results demonstrated the high accuracy of the models we developed.

To further evaluate the model, the relative error (RE) of the determined values was calculated. In general, the RE is less than 5% and indicates that the determined values are very close to the values in the reference. As can be seen from Figure 5, the RE of polysaccharides, water-soluble extract, alcohol-soluble extract and the sum of five triterpenoids is 4.055%, 3.821%, 4.344% and 3.744%, respectively. Therefore, it can be concluded that four models have high levels of accuracy and are capable of the fast quantitative analysis of PC chemical components.

## 4. Conclusions

The present study combined NIR spectroscopy with chemometrics, including RFM and PLSR, and developed models to determine PC cultivation regions and quality parameters represented by water-soluble extract, alcohol-soluble extract, polysaccharides and the sum of five triterpenoids. We selected optimal pretreatment method and chemometric method to optimize sample classification and generate quantitative models. Using the classification model, we achieved a classification accuracy of 92.59% for the validation group, with only two out of 27 samples incorrectly classified. In the meantime, we developed and optimized the PLSR model that was independent of the calibration or prediction set, with *R*^2^ greater than 0.9 and relative error smaller than 5% for most of the samples. The results verified the feasibility of this method in measuring the content of PC chemical components.

PC is a medicinal and nutritional material commonly used in functional food, healthy snacks and drinks. In recent years, demand for PC has increased continuously, leading to the increasing popularity of PC cultivation. Many factors of cultivation environment affect the quality of PC and cause discrepancies in the quality of products from different regions. NIR spectroscopy is an electromagnetic wave that is mainly the combination of overtone bands from hydrogen-containing function groups, such as C–H, N–H, O–H and their combination of both C–O stretching and O–H deformation. These absorptions contain rich composition information, making NIR spectroscopy very suitable for the measurement of hydrocarbon organic compounds in PC polysaccharides, water-soluble extract, alcohol-soluble extract and triterpene acids. In this study, we used the NIR method, which has multiple advantages over traditional methods, to rapidly and accurately evaluate the quality of PC. The method can be applied to general screening and rapid quality estimation, which, together with more samples in the future, will further improve the accuracy of the models we developed.

## Figures and Tables

**Figure 1 foods-11-00892-f001:**
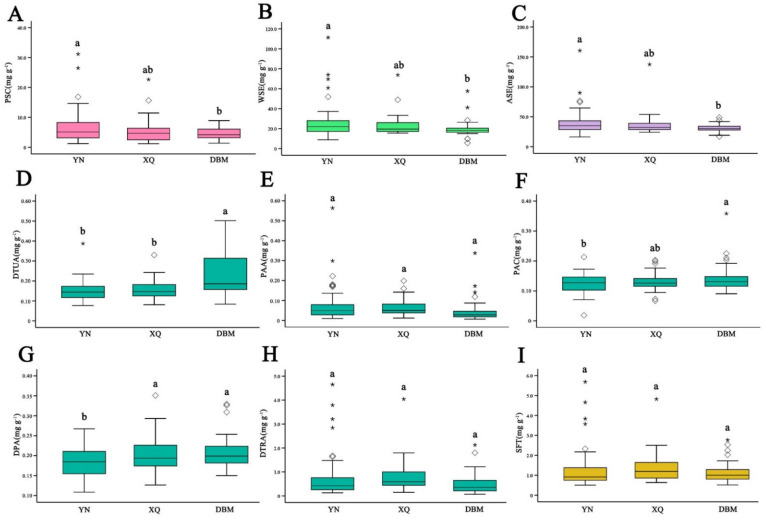
Box and whisker plot of polysaccharides (PSC, **A**), water-soluble extract (WSE, **B**), alcohol-soluble extract (ASE, **C**), triterpene acids DTUA (**D**), PAA (**E**), PAC (**F**), DPA (**G**), DTRA (**H**) and sum of five triterpenoids (SFT, **I**) in cultivation regions of PC. Data are mean values ± SD (significant differences between the letters). The boxes are bounded by the 25% and 75% quartiles, with the median inside, whereas the extreme levels correspond to 5% and 95% percentiles, the rhombuses and stars represent the atypical values.

**Figure 2 foods-11-00892-f002:**
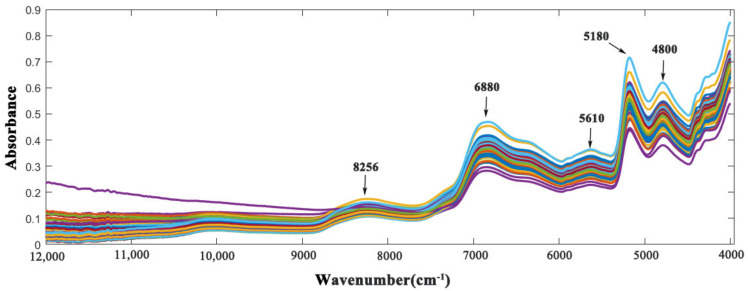
NIR spectra of the 138 PC samples without pretreatments.

**Figure 3 foods-11-00892-f003:**
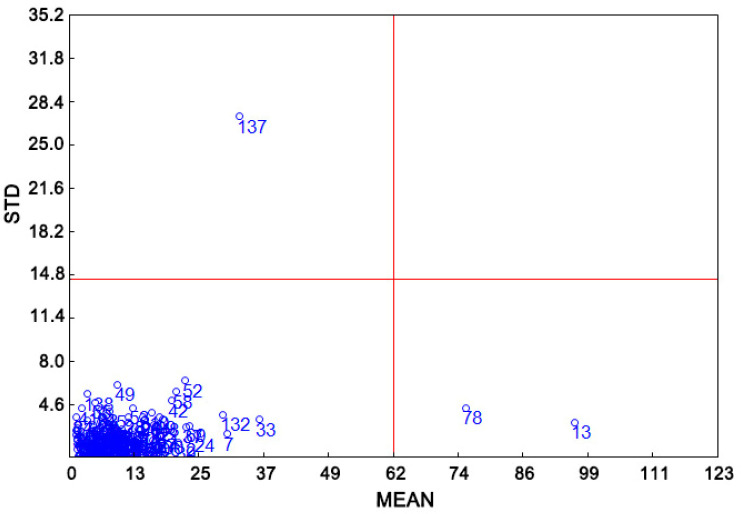
Outlier identification by using Monte Carlo cross-validation method.

**Figure 4 foods-11-00892-f004:**
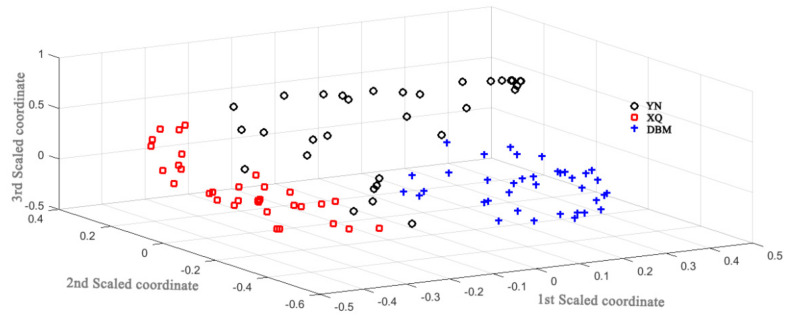
The NIR-based classification plot of PC for three main producing regions.

**Figure 5 foods-11-00892-f005:**
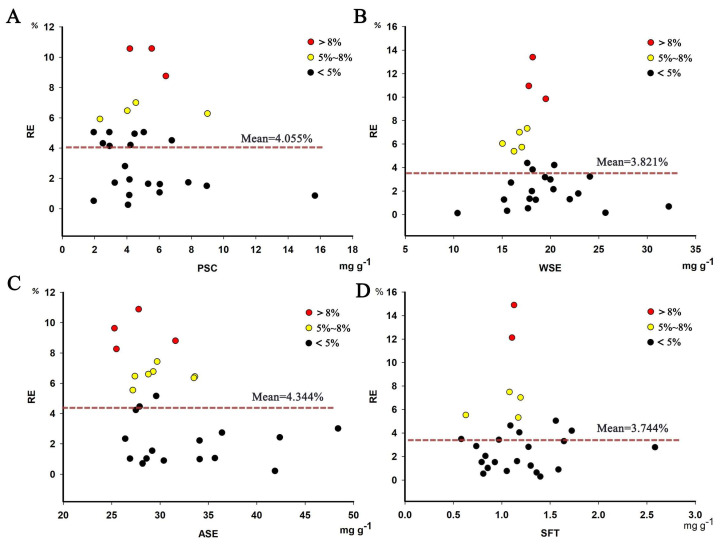
The relative error (RE) of polysaccharides (PSC), water-soluble extract (WSE), alcohol-soluble extract (ASE) and sum of five triterpenoids (SFT). (**A**) polysaccharides; (**B**) water-soluble extract; (**C**) alcohol-soluble extract; (**D**) sum of five triterpenoids.

**Table 1 foods-11-00892-t001:** Linear regression of the five triterpene acids.

Compounds	Range (μg mL^−1^)	Regression Equation ^a^	*R^2^*	LOD ^b^ (μg mL^−1^)	LOQ ^c^ (μg mL^−1^)
DTUA	0.81~40.60	*y* = 16,985,351.48000 *x* + 4404.85745	0.996	0.209	0.696
PAA	0.40~20.00	*y* = 14,178,282.95000 *x* − 1464.46374	0.998	0.186	0.619
PAC	0.41~20.40	*y* = 16,709,116.59000 *x* − 399.56329	0.999	0.166	0.554
DPA	0.79~39.60	*y* = 15,817,965.09000 *x* + 518.67483	0.998	0.236	0.788
DTRA	1.21~60.40	*y* = 18,347,795.59000 *x* + 9040.36403	0.994	0.157	0.525

^a^, *x*—concentration of the analyte (mg mL^−1^), *y*—peak value; ^b^, LOD—limit of detection; ^c^, LOQ—limit of quantification.

**Table 2 foods-11-00892-t002:** Active contents of PC samples.

Regions		Yunan Region	Xiangqian Region	Dabie Mountains Region
Number of Samples		58	28	52
PSC (mg g^−1^)	Min	1.24	1.20	1.40
	Max	31.16	22.66	8.96
	Mean	6.81	5.67	4.59
	SD	5.62	4.59	1.89
WSE (mg g^−1^)	Min	8.80	15.52	5.68
	Max	73.95	73.76	57.58
	Mean	25.22	23.95	19.57
	SD	12.81	12.18	7.40
ASE (mg g^−1^)	Min	16.29	24.20	17.01
	Max	90.00	53.88	48.70
	Mean	37.85	33.60	31.24
	SD	14.31	7.53	5.83
DTUA (mg g^−1^)	Min	0.08	0.08	0.08
	Max	0.39	0.33	0.50
	Mean	0.15	0.16	0.24
	SD	0.05	0.05	0.10
PAA (mg g^−1^)	Min	0.01	0.01	0.01
	Max	0.30	0.20	0.34
	Mean	0.06	0.07	0.05
	SD	0.06	0.05	0.05
PAC (mg g^−1^)	Min	0.02	0.07	0.09
	Max	0.21	0.20	0.36
	Mean	0.13	0.13	0.14
	SD	0.03	0.03	0.04
DPA (mg g^−1^)	Min	0.11	0.13	0.15
	Max	0.27	0.35	0.33
	Mean	0.18	0.21	0.21
	SD	0.04	0.05	0.04
DTRA (mg g^−1^)	Min	0.13	0.15	0.07
	Max	4.65	4.04	2.12
	Mean	0.73	0.80	0.50
	SD	0.89	0.73	0.41
SFT (mg g^−1^)	Min	0.51	0.64	0.52
	Max	5.68	4.82	2.77
	Mean	1.26	1.36	1.13
	SD	1.00	0.80	0.47

PSC—polysaccharides; WSE—water-soluble extract; ASE—alcohol-soluble extract; DTUA—Dehydrotumulosic acid; PAA—poricoic acid A; PAC—polyporenic acid C; DPA—dehydropachymic acid; DTRA—Dehydrotrametenolic acid; SFT—sum of the five triterpenoids; SD—standard deviation.

**Table 3 foods-11-00892-t003:** Comparison between reference values and the values determined by NIR spectroscopy and RFM in validation samples.

Sample	Reference Values	NIR Values	Results
S16	YN ^a^	YN	Correct
S19	YN	YN	Correct
S22	YN	YN	Correct
S25	YN	YN	Correct
S35	YN	YN	Correct
S50	YN	YN	Correct
S57	DBM ^b^	DBM	Correct
S64	DBM	DBM	Correct
S65	DBM	DBM	Correct
S67	DBM	DBM	Correct
S68	DBM	DBM	Correct
S69	DBM	DBM	Correct
S73	DBM	DBM	Correct
S76	DBM	DBM	Correct
S80	DBM	DBM	Correct
S83	XQ ^c^	XQ	Correct
S88	XQ	XQ	Correct
S92	XQ	DBM	False
S93	DBM	DBM	Correct
S101	DBM	DBM	Correct
S102	DBM	DBM	Correct
S109	DBM	DBM	Correct
S112	DBM	DBM	Correct
S121	YN	YN	Correct
S128	DBM	DBM	Correct
S133	YN	YN	Correct
S136	XQ	DBM	False

^a^, YN—Yunnan region; ^b^, DBM—Dabie Mountains region; ^c^, XQ—Xiangqian region.

**Table 4 foods-11-00892-t004:** Range of values obtained by the reference methods for PC samples in the calibration and validation sample groups.

Indexes	Calibration				Validation			
	N	Range	Mean	SD	N	Range	Mean	SD
PSC	108	1.20–31.16	5.89	4.84	27	1.95–8.90	4.97	2.02
WSE	108	5.68–73.95	23.31	11.08	27	10.40–25.69	18.28	3.10
ASE	108	16.29–90.00	35.27	11.73	27	25.29–42.38	30.77	4.51
SFT	108	0.51–5.68	1.25	0.87	27	0.58–2.53	1.12	0.40

PSC—polysaccharide content; WSE—water-soluble extract content; ASE—alcohol-soluble extract content; SFT—sum of the five triterpenoids; SD—standard deviation.

**Table 5 foods-11-00892-t005:** Results obtained by using different methods for signal pretreatment.

Indexes	Evaluations ^a^	Raw	MSC	SNV	Smooth	Smooth + SNV	Smooth + MSC	SG-1D
PSC	RMSEC	0.122	0.107	0.114	0.112	0.101	0.093	**0.056**
	*R* ^2^ _cal_	0.855	0.888	0.872	0.879	0.901	0.915	**0.987**
	RMSEP	0.179	0.165	0.188	0.108	0.138	0.124	**0.079**
	*R* ^2^ _pre_	0.739	0.790	0.701	0.863	0.805	0.859	**0.965**
WSE	RMSEC	2.802	2.924	2.455	2.343	3.116	2.489	**1.083**
	*R* ^2^ _cal_	0.824	0.822	0.864	0.877	0.782	0.861	**0.978**
	RMSEP	3.315	3.090	3.174	3.467	3.610	3.516	**1.934**
	*R* ^2^ _pre_	0.758	0.808	0.768	0.725	0.696	0.710	**0.962**
ASE	RMSEC	2.996	2.701	**1.502**	2.066	2.489	3.749	3.512
	*R* ^2^ _cal_	0.910	0.920	**0.972**	0.946	0.911	0.897	0.909
	RMSEP	2.772	3.763	**1.926**	2.968	4.285	3.894	3.763
	*R* ^2^ _pre_	0.910	0.862	**0.940**	0.918	0.836	0.873	0.862
SFT	RMSEC	0.298	0.228	**0.165**	0.225	0.225	0.229	0.197
	*R* ^2^ _cal_	0.862	0.919	**0.975**	0.921	0.921	0.919	0.940
	RMSEP	0.315	0.279	**0.191**	0.298	0.283	0.270	0.259
	*R* ^2^ _pre_	0.846	0.880	**0.961**	0.862	0.876	0.887	0.896

PSC—polysaccharide content; WSE—water-soluble extract content; ASE—alcohol-soluble extract content; SFT—the sum content of five triterpene acids; ^a^—the smaller the RMSEC, RMSEP value and the closer the *R*^2^_cal_, *R*^2^_pre_ value to 1, the better the model.

## Data Availability

The data presented in this study are available in the article.

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
