# Peer review of "Determination of Cultivation Regions and Quality Parameters of Poria cocos by Near-Infrared Spectroscopy and Chemometrics"

_foods, 2022, doi:10.3390/foods11060892_

Round 1
Reviewer 1 Report
The authors did a good work from an experimental point of view, and I recommend the article for publication after some minor revisions.
More specific:
L32: ‘’Poria cocos’’ and ‘’P. cocos’’ in italics.
L121: What was the volume of the solvent?
L162: The model and the company of Ultrasonicator are missing. The methodology with some details is also missing.
Reviewer 2 Report
-LOD and LOQ values were not calculated
-Figure 3 can be deduced as outliers were explained at lines 267-270.
-What is the method of PLSR? (second derivative etc.)
-What are the prediction factors?
-AT Table 6 top section; 'NIR val-ues' should be as 'NIR values'.
-The discussion of the molecular properties for polysaccharides, (B) water-soluble extract, (C) ethanol-soluble 326 extract, and (D) sum of the five triterpenoids, were not enough. It is very important for using the IR. The prediction results should be combined with the chemometric results. It should be summaries in the discussion section also.
-Chemicals bonds of the predicted molecules should be written detailed.
-Range of values obtained by the reference methods for PC samples in the calibration and 299 validation sample groups. Range of the calibration and validation sets for WSE as 5.68-73.95 and 10.40-25.69 , for ASE as 16.29-90.00 and 25.29-42.38, for PSC 1.20-31.16 and 1.95-8.90, for SFT 0.51-5.68 and 0.58-2.53. What is the selection method of the validation and calibration groups? The ranges for the WSE, ASE, and PSC were not suitable for each other between validation and calibration groups. This is not good for the prediction models.
Reviewer 3 Report
My comments on the content and form of the manuscript
Lines 72 and 73: Several review manuscripts deal specifically with fungi and their NIR analysis. References to other agricultural products are redundant.
I suggest the inclusion of the following manuscripts in the introduction section:
- Manley, M. Near-infrared spectroscopy and hyperspectral imaging: Non-destructive analysis of biological materials. Chem. Soc. Rev. 2014, 43, 8200–8214 https://doi.org/10.1039/C4CS00062E
- Meenu, M., & Xu, B. Application of vibrational spectroscopy for classification, authentication and quality analysis of mushroom: A concise review. Food Chemistry, 2019, 289, 545–557. https://doi.org/10.1016/j.foodc hem.2019.03.091
- Chen, Y., Xie, M., Zhang, H., Wang, Y., Nie, S., & Li, C. Quantification of total polysaccharides and triterpenoids in Ganoderma lucidum and Ganoderma atrum by near infrared spectroscopy and chemometrics. Food Chemistry, 2012, 135(1), 268–275. https://doi.org/10.1016/j.foodchem.2012.04.089
Application of Near-Infrared Spectroscopy to Investigate Some Endogenic Properties of Pleurotus ostreatus Cultivars. Sensors 2020, 20, 6632
https://doi.org/10.3390/s20226632
Line 92: water soluble and alcohol soluble extract - specifically what components do you mean when you use these terms?
Line 128: why is the multiplication by 2 in the formula?
Line 149: if the calibration is set up for a pure glucose solution, what is the reason for the calibration line not starting from zero, i.e. why is it not of type y=ax ? What causes the "+b" discrepancy?
Line 163: DTUA, PAA, PAC, DPA, DTRA - abbreviations not named in the manuscript
Previously it was wrote that 91 bioactive triterpenes were identified (line 66). Why were these five selected?
Line 169: FT-NIR is not a spectrophotometer but a spectrometer
Line 201: What does the abbreviation RMSER stand for?
Lines 215-223: it is advisable not to abbreviate the cultivation sites, it is easier to understand if you can see the name of the site. The data given will appear later in Table 2. It is advisable to provide the tabular data earlier than the evaluation and to highlight in bold type in the table the data considered particularly important.
Lines 231-236: abbreviations are not resolved
Line 242: In Table 2, what does the standard deviation refer to? For example, the water soluble extract content of samples from the same growing area ranges from 8.798 to 73.949 mg/g. What does the standard deviation refer to?
The data in Table 2 should be revised, the precision of 3 decimal places is exaggerated (especially for values greater than 10)
Explanation of abbreviations is given at the end of Table 2. But the order shown in the table and the order of the abbreviations are confused.
This needs to be reconciled - as the parameters come in the table, so should they be written at the end of the table.
Chapter 3.2 : missing references
Figure 2 is incorrect. When plotted as a function of wavenumber, we always move from the higher wavenumber to the lower one. The mirror image of the spectra is correct.
Chapter 3.3: 3 outliers found. What was the reason for this?
Line 283: In Figure 4, the name of x, y, and z axes are missing
Line 300: the units of measurement are missing from Table 4. The order of the parameters in the table and the explanation of the abbreviations are confused. This needs to be harmonised.
Same error in Table 5.
I suggest that the parameters (and the explanation of the abbreviations) in Tables 2, 4 and 5 should be in the same order.
Please add the evaluation ranges to Table 5. Only the range of measurement is given in the text.
Which variable selection methods were used for each component?
Line 307: typo, correct Savitzky-Golay
Lines 313-316: These data have already been reported in Table 5, why do they need to be rewritten?
Lines 323-324: Figure 5 is incorrect. In multivariate regression extrapolation is forbidden! The maximum x value is the maximum reference value. And of curse the last y value is the value, what is passt to the maximum reference value. The regression lines in all 4 figures are incorrect for this reason.
However, I consider this figure unnecessary, as the data in Table 5 speak for themselves. Figure 5 provides no additional information.
I propose a figure instead of Table 6 because it is more illustrative. For example, one where the x-axis is the reference value, the y-axis with +/- sign is the relative error
Round 2
Reviewer 3 Report
The authors have corrected the problems I raised.
I accept the manuscript as it stands.